# Lightweight UAV Landing Model Based on Visual Positioning

**DOI:** 10.3390/s25030884

**Published:** 2025-01-31

**Authors:** Ning Zhang, Junnan Tan, Kaichun Yan, Sang Feng

**Affiliations:** School of Electromechanical Engineering, Guangdong University of Technology, Guangzhou 510006, China13536066055@163.com (J.T.);

**Keywords:** aerial photograph, landing sign, YOLOv8, lightweight, bidirectional characteristic pyramid, BiFPN structure

## Abstract

In order to enhance the precision of UAV (unmanned aerial vehicle) landings and realize the convenient and rapid deployment of the model to the mobile terminal, this study proposes a Land-YOLO lightweight UAV-guided landing algorithm based on the YOLOv8 n model. Firstly, GhostConv replaces standard convolutions in the backbone network, leveraging existing feature maps to create additional “ghost” feature maps via low-cost linear transformations, thereby lightening the network structure. Additionally, the CSP structure of the neck network is enhanced by incorporating the PartialConv structure. This integration allows for the transmission of certain channel characteristics through identity mapping, effectively reducing both the number of parameters and the computational load of the model. Finally, the bidirectional feature pyramid network (BiFPN) module is introduced, and the accuracy and average accuracy of the model recognition landing mark are improved through the bidirectional feature fusion and weighted fusion mechanism. The experimental results show that for the landing-sign data sets collected in real and virtual environments, the Land-YOLO algorithm in this paper is 1.4% higher in precision and 0.91% higher in mAP0.5 than the original YOLOv8n baseline, which can meet the detection requirements of landing signs. The model’s memory usage and floating-point operations per second (FLOPs) have been reduced by 42.8% and 32.4%, respectively. This makes it more suitable for deployment on the mobile terminal of a UAV.

## 1. Introduction

With the increasing demand for drones in the military, agricultural, transportation, and civil fields, the demand for fully autonomous flight drones is becoming more and more urgent. In these specific flight-application scenarios, drones need to have a high degree of autonomous flight capability to successfully perform diverse tasks, and ultimately achieve autonomous landing and effective recovery [1,2].

Autonomous landing technology is the cornerstone of the UAV’s critical landing phase during flight. Its main goal is to guarantee the UAV’s safe and precise landing within the designated area, thereby enhancing its overall autonomous flight capabilities. At present, the autonomous landing technology of UAV mainly relies on its own navigation and positioning system and target detection algorithm for landing positioning. Due to the low positioning accuracy of GPS and the instability of satellite signals, it is easy to face problems such as excessive deviation or even landing failure during landing. With the development of computer-vision technology, coupled with the advantages of low cost, strong signal anti-interference ability, and rich image information, vision-based navigation technology has become a mainstream choice for UAV autonomous landing [3,4]. However, vision-based landing can pose challenges, such as false detections and missed detections, for the target detection model. At the same time, the trained model is difficult to deploy to the terminal due to the large amount of parameters.

Current research is focused on how to enhance the precision of UAV landings. Many scholars have carried out relevant research. Borowczyk et al. [5] introduced a method for autonomous UAV landing that relies on the PID (Proportional–Integral–Derivative) algorithm. Using Kalman filter technology, the inertial navigation system (INS) signal loaded on the UAV is deeply fused with the information obtained by visual detection of the landing mark, so as to obtain the spatial position relationship between the UAV and the landing mark. Lin et al. [6] proposed a PBVS (Position-based Visual Servo)-based autonomous landing algorithm for UAVs. According to the underactuated characteristics of the quadrotor, the control strategy comprises two modules: an outer-loop position control module and an inner-loop attitude control module. These modules work together to ensure the stability of the landing process and to effectively manage complex, nonlinear landing systems. Li et al. [7] introduced a novel feature extraction framework tailored to address the challenges of a small field of view and significant image-scale variations in unmanned helicopters. By integrating the SSD and KCF algorithms for mutual correction and leveraging the strengths of deep learning, they ensured the framework met the demands of autonomous UAV landing tasks. Lee B et al. [8] proposed a landing algorithm that combines visual inspection technology with a nonlinear controller. Even if a time delay is generated during the deep learning target-detection process, the algorithm can still achieve real-time autonomous flight control and ensure the continuity and safety of the landing process. Guan et al. [9] introduced a novel autonomous landing controller, leveraging the backstepping control algorithm. This controller employs a performance-constraint guidance law to ensure that the trajectory tracking error remains within an acceptable range. Yang et al. [10] proposed the SCRDet detection model and designed a sampling fusion network to improve the sensitivity of small targets in aerial images. Li et al. [11] proposed DMNet to generate a density map and learn the proportion information based on the density intensity, which improves the detection performance of small targets. The above improvements can improve the problem of low accuracy of UAV autonomous landing to a certain extent, but the algorithm model used is too large, resulting in poor real-time performance and large computational consumption.

In this regard, this paper will create the corresponding landing-sign data set, train the detection model, and make improvements to achieve a lightweight model. The specific improvement scheme is as follows: use GhostConv instead of Conv in the model structure to reduce the size of the weight file; the CSP structure is improved, and the redundancy of feature map is reduced by channel pruning. The BifFPN module is added to enhance the feature-fusion ability and improve the network ’s ability to capture small targets. Finally, the trained model is simulated and verified on the AirSim platform.

## 2. Overview of YOLOv8

Currently available in the market are two primary types of deep learning-based target detectors: one-stage and two-stage detectors. The mainstream of the two-stage target detector is R-CNN [12,13], etc. Its working principle includes the production of the target box, the extraction and coding of the feature vector, and the category and position regression of the target object. The one-stage object detection algorithm has SSD [14], YOLO series [15,16,17], etc. Its core abandons the stage of generating the proposal box, and classifies each region of interest as the background or the target object. Hence, the first-stage detector boasts a quicker inference speed, making it ideal for real-time object detection applications. For scenes that require fast target detection, a one-stage target-detection algorithm is usually used.

YOLOv8 is the latest YOLO series model developed by Ultralytics. It introduces a new structure based on YOLOv5, which further improves performance and scalability, and achieves high accuracy while maintaining high detection speed. Figure 1 displays the YOLOv8 model framework, consisting primarily of the input, backbone, neck, and head components. The input section is primarily tasked with processing and augmenting raw data, while the backbone component is responsible for extracting features from the input image. The shallow network is responsible for extracting the underlying features such as the target edge, and the deep network is responsible for constructing high-level semantic information. The neck processes the features extracted from the backbone network, and uses the PAN and FPN network structures to make the features more fully fused through top-down and bottom-up cross-layer connections. In the final head section, the prevalent decoupling head architecture is employed to segregate the classification and detection tasks, and the positive and negative samples are determined according to the scores weighted by the scores of classification and regression, which effectively enhances the model’s performance. The anchor has also been abandoned, and a more flexible and precise anchor-free variant is employed to accommodate targets of varying sizes and shapes.YOLOv8 has undergone several versions of optimization. According to the width and depth of the network, it is divided into YOLOv8n, YOLOv8s, YOLOv8m, YOLOv8, and YOLOv8x. The YOLOv8n model is one of the most basic networks in the YOLOv8 system. It possesses high accuracy, along with swift detection speed. Therefore, YOLOv8n is more suitable for the deployment of UAV detection algorithms.

## 3. Improvement of YOLOv8n Algorithm

The existing convolutional neural network (CNN) based on deep learning exhibits high detection accuracy; however, it is characterized by high computational complexity, large size, and slow detection speed.

This study introduces an enhanced target-detection algorithm, Land-YOLO, which is built upon YOLOv8n. It is mainly optimized and improved from the following three aspects to achieve higher detection performance and lower computational cost. Firstly, to decrease the computational complexity of convolution, the network structure employs GhostConv in place of traditional convolution, thereby reducing the size of the weight file and enhancing the model’s lightness. Secondly, the CSP structure is refined by incorporating the Pconv concept, leading to the proposal of a novel structure, CSPPC. This reduces feature map redundancy through channel pruning and enhances the detection rate. Lastly, the BifFPN module is integrated to bolster feature-fusion capabilities, thereby improving the accuracy and robustness of target detection. This addresses the accuracy loss associated with the lightweight process. The improved structure is shown in Figure 2. Among them, the color mark is an improved network module.

### 3.1. GhostConv

In YOLOv8n, the convolutional neural network captures rich feature information by stacking a large number of convolutional layers. However, this approach also brings a huge amount of computation and parameters, which is not conducive to deployment on embedded devices or mobile terminals.

The core idea of GhostConv [18] is to use the existing feature maps to generate more Ghost feature maps through low-cost linear transformation, thereby improving the computational efficiency of the network. The standard convolution layer typically includes regular convolution, batch normalization, and the ReLU activation function. Alternatively, the GhostConv structure, depicted in Figure 3, offers a different approach. First, the input image is compressed by using ordinary convolution, batch normalization, and an activation function, Relu, to generate some intrinsic feature maps. At the same time, the feature map is applied to a series of linear operations to obtain φk and more feature maps, add features, and splice the feature maps obtained in the previous two steps to obtain the final output feature map.

The speedup formula can directly provide the improvement ratio of the calculation speed using GhostConv relative to the traditional convolution. rs represents the ratio of the amount of calculation through the ordinary convolution to the amount of calculation through the Ghost module. The derivation formula is as follows:(1)$$\begin{array}{cc} r_{s} & =\frac{n\cdot h\cdot w\cdot c\cdot k\cdot k}{\frac{n}{s}\cdot h\cdot w\cdot c\cdot k\cdot k+\left(s-1\right)\cdot \frac{n}{s}\cdot h\cdot w\cdot d\cdot d}\\ & =\frac{c\cdot k\cdot k}{\frac{1}{s}\cdot c\cdot k\cdot k+\frac{s-1}{s}\cdot d\cdot d}\approx \frac{s\cdot c}{s+c-1}\;\approx s \end{array}$$

Comparing the parameters of the traditional convolution with the parameters of the Ghost convolution, the formula of the parameter compression ratio can be obtained. rc represents the ratio of the number of ordinary convolution parameters to the number of parameters passing through the Ghost module. The derivation formula is as follows:(2)$$r_{c}=\frac{n\cdot c\cdot k\cdot k}{\frac{n}{s}\cdot c\cdot k\cdot k+\left(s-1\right)\cdot \frac{n}{s}\cdot d\cdot d}\approx \frac{s\cdot c}{s+c-1}\approx s$$

Among them, ‘*c*’ denotes the input channel, ‘*n*’ represents the output channel, and ‘*h*’ and ‘w’ signify the height and width of the output feature map, respectively. The variable ‘*k*’ indicates the size of the traditional convolution kernel, ‘*d*’ refers to the size of the linear transformation convolution kernel, and ‘*s*’ represents the number of transformations.

From the above formulas, it can be seen how the GhostConv module increases the number of channels by splicing the basic features and cheap features, and avoids a large number of multiplication and addition operations in traditional convolution, so as to achieve a significant reduction in the amount of calculation and parameters.

Using GhostConv instead of the traditional convolution in YOLOv8 can effectively reduce the amount of calculation required, maximize the use of available computing and memory resources, make the improved model easier to deploy in the UAV terminal, and enhance the real-time performance of the algorithm target detection.

### 3.2. Multi-Scale Feature-Fusion BiFPN

The C2f module of the YOLOv8 neck network contains multiple convolution layers and residual connections, which can process and fuse the stitched features. However, from the perspective of a UAV, due to small targets occupying fewer pixels in the image, some information is easily ignored in the feature extraction stage, resulting in the loss of key target information.

To address this limitation and enhance the model’s performance in detecting small targets within aerial images, this paper adds a BiFPN—bidirectional feature pyramid network—to the neck network, and the structure is shown in Figure 4. FPN mainly relies on the top-down path to fuse features, and the bidirectional connection of BiFPN provides a richer way of feature fusion.

BiFPN [19,20] contains three key components: a bottom-up feature-fusion path, a top-down feature enhancement path, and a horizontal connection. The feature-fusion path fuses the shallow features from the backbone network layer by layer to obtain richer semantic information. The feature enhancement path transmits the high-level features from the head network layer by layer downward, and fuses them with the bottom-up features to enhance the detail information of the features. At the same time, at each stage, the bottom-up features are horizontally connected with the top-down features to ensure information sharing between features at different scales.

In BiFPN, each two-way path (top-down and bottom-up) is considered as a separate feature network layer, which can then be repeated multiple times to facilitate higher-level feature fusion. As a result, a simplified two-way network is created, which boosts the network’s feature-fusion capabilities, enabling it to utilize information from different scales more effectively. This, in turn, enhances the network’s ability to detect small targets.

### 3.3. CSP-CSPPC Combined with PConv

The CSP structure in YOLOv8 is a deep neural network structure. The core of the structure is the divide-and-conquer strategy. It improves the expression ability and generalization ability of the model by dividing the input features into two parts and performing cross-connection and feature fusion between the two parts.

In this paper, lightweight PartialConv [21] is used to improve CSP, and a new structure for CSPPC is proposed to improve the detection rate and control over-fitting. As shown in Figure 5, first of all, the input information is used to expand the number of channels by 1 × 1 convolution to ensure the dimension matching of input and output and maintain the consistency of the network. The extended feature map is divided into two parts. After the PartialConv processing of P1, it is spliced with P2 in the channel dimension. The spliced feature map reduces the number of channels through another 1 × 1 convolution to obtain the final output.

The CSPPC structure reduces the amount of network parameters by introducing PartialConv to reduce computing and memory access. PartialConv is a convolution with high-speed reasoning ability, which can effectively reduce the redundancy of feature maps and improve the computational efficiency of the network. The PartialConv structure is shown in Figure 6, which achieves fast and efficient operation by applying filters on only a few input channels while retaining other channels unchanged. Compared with conventional convolution, PConv reduces FLOPs, and has higher FLOPs than deep/block convolution, which reduces the amount of computation that must be performed and reduces memory access. Using the improved CSSPC structure, the detection rate and running rate of the algorithm can be improved.

## 4. Experimental Preparation 

### 4.1. Data Sets

In previous studies, researchers have designed a large number of cooperative-based landing signs as autonomous landing areas. Commonly used shapes include: ‘T-shaped’, ‘H-shaped’, ‘round’, ‘square’, ‘red indicator light’, and a combination of various forms [22,23], as shown in Figure 7.

In view of the fact that there is no public authoritative landing sign data set in the current research, this paper selects the commonly used ‘H-shaped’ variant as the landing sign, and creates the data set and outputs it to match the format of the YOLOv8 network model. Considering that the UAV needs to respond to the apron logo in the simulation, 1000 overlapping pictures of the apron were collected, including different illuminations, different perspectives, different sizes, etc. In addition, in order to enrich the data set, this study introduces a virtual cylinder in the scene and draws a map of the apron. In UE4, the cylinder and the map are combined to realize the construction of the virtual apron. Then, 400 virtual apron images are obtained through the flight records of the simulation platform.

The above images are translated, rotated, scaled, flipped, and cropped to enhance the data, and a data set is constructed. There are 3293 images in total, and the data set is shown in Figure 8. Among them, the training set contains 2403 images, aiming to fully train the model; in the validation set, 594 images were selected for performance verification during model tuning. The test set contains 296 images to evaluate the generalization ability of the model on unseen data.

### 4.2. Experimental Environment and Parameters

The CPU used in this paper is 6-core E5-2680v4, and the GPU is RTX3080, with 12.6 G memory. The software environment used is: Ubuntu20.04, Python3.10,12, CUDA11, PyTorch2.0.1 deep learning framework. An optimizer is an algorithm for updating model parameters. Different optimizers have different update strategies and convergence speeds. Momentum is used to accelerate the optimization of SGD (stochastic gradient descent) in the correlation direction and suppress oscillations. The lr0 (learning rate) determines the step size of the parameter update in the optimization process. Epoch refers to the number of times the entire training data set is traversed. A sufficient number of epochs ensures that the model fully learns the data features. Batch size refers to the number of data samples used to train the model in one iteration. The above parameters are set as shown in Table 1.

### 4.3. Model-Evaluation Indices

This paper mainly uses the following evaluation indicators to evaluate the results of the algorithm [24].

Precision (P) is a measure of the proportion of positive instances predicted by the model that are actually positive. This index helps to evaluate the accuracy of the model prediction results.

The recall rate (recall, R) reflects the proportion of the model successfully predicted as a positive class in all actual positive-class samples. It serves as a metric to evaluate the model’s capability to identify and encompass all positive samples.

MeanAveragePrecision (mAP) is a commonly used evaluation index, which is especially suitable for the model performance of target-detection tasks. It is the mean of multiple category-specific average precisions (APs). Specifically, mAP @ 0.5 signifies the average of APs for all categories when the Intersection over Union (IoU) threshold is set to 0.5.

In order to comprehensively evaluate the performance of the model, the parameters and computational complexity (GFLOPs) are usually considered comprehensively.

### 4.4. Ablation Experiment

To assess the efficacy of each proposed improvement method in this paper, we utilize YOLOv8n as the baseline model and evaluate the detection outcomes of various enhanced techniques. Data statistics are shown in Figure 9 and Table 2. These experiments were tested for a single improvement and a combination of multiple improvements.

As evident from the ablation experiment table, substituting GhostConv convolution for standard convolution in the baseline model YOLOv8n results in the number of floating-point operations and the memory usage of the model decreasing significantly, by 0.55 × 109 and 0.28 M, respectively, which is convenient for subsequent terminal deployment.

The BiFPN module is used to reconstruct the neck network structure. The two-way connection mechanism in BiFPN allows for information to be transmitted in both directions between different resolution levels, which increases precision, recall, and mAP50 by 1.07%, 0.17%, and 0.8%, respectively. Due to the increased complexity of the reconstructed neck network, GFLOPs have increased.

The improved CSP reduces the floating-point operation by calculating on fewer channels, and extracts richer features through cross-linking and feature fusion to improve the generalization ability. It can be seen from the table that the improved model is significantly lower than the baseline model GFLOPs, and precision and mAP50 are also increased by 0.3% and 0.25%, respectively.

For a comprehensive comparison, the Land-YOLO algorithm proposed in this paper has greatly improved the evaluation indexes of YOLOv8n. Precision, recall, and mAP50 have increased by 1.4%, 0.25%, and 0.91%, respectively, while Parmas and GFLOPs have decreased by 42.8% and 32.4%, respectively.

### 4.5. Comparative Experiment

To further validate the effectiveness of the improved algorithm, the improved algorithm Land-YOLO is compared with the current mainstream target-detection algorithms YOLOv10n, YOLOv8n, YOLOv5n, YOLOv7—tiny, and Faster-RCNN using the same experimental equipment and training strategy, without changing the experimental parameters and data sets. The experimental results are shown in Table 3.

As demonstrated in Table 3, the detection performance metrics of the proposed algorithm surpass those of current mainstream algorithms significantly, further validating the efficacy of the improved algorithm presented in this paper.

### 4.6. Visualization Analysis

The experimental data of the Land-YOLO algorithm is shown in Figure 10. Its training and verification process contains three loss functions—box_loss, cls_loss, and dfl_loss—which have gradually converged smoothly around 150 rounds. The four index evaluation indexes, precision, recall, mAP50, and mAP50-95, have also reached the convergence state, and all of them have reached more than 95%, which shows the feasibility of the algorithm from the data level. In order to show the detection effects of the algorithm designed in this paper, the following is a visual comparison of the detection results of the original algorithm and the improved algorithm. The comparison results are shown in Figure 11.

Four special scenes are selected in the figure, which are a tilted-view scene, a multi-target scene, a blurred small-target scene, and a low-illumination scene. In the first scene, the original algorithm can identify the landing mark, but the confidence threshold is low, while the improved algorithm improves the confidence of the landing mark from this perspective by 23.8%. In the second scenario, the original algorithm has false detection and detection box overlap, and Land-YOLO can improve this situation well. In the third scene, the original algorithm cannot recognize the landing mark in the image because it fails to extract the image features well, while the improved algorithm can effectively catch small targets by fusing the features extracted by the two-way feature pyramid network. In the fourth scenario, both algorithms can recognize the landing mark and the confidence level reaches more than 80%, which is due to the integrity and similarity of the data set.

In summary, under the premise of ensuring high precision, the Land-YOLO model proposed in this study has the smallest memory footprint, a low number of floating-point operations per second, and the best comprehensive performance in landing-sign recognition. 

### 4.7. Visual Simulation of UAV Flight Based on AirSim

AirSim is an open-source, multi-platform unmanned aerial vehicle (UAV) simulator built on Unreal Engine, capable of simulating UAV behavior across diverse scenarios [29]. In this paper, AirSim based on UE4 is selected as the UAV simulator for development. AirSim can be applied to any scene of UE4 and Unity. After building the joint simulation platform, the positioning information and pose estimation information during the landing process of the UAV are updated by deploying the improved YOLOv8 visual guidance landing algorithm in the AirSim interface. The landing icon is deployed in the UE4 virtual environment, the UAV flight plan is set to simulate the landing process of the multi-rotor UAV based on visual guidance, and the virtual landing experiment is carried out. As shown in Figure 12, the flight–landing test scenarios of the four-rotor UAV is simulated in a forest environment (a,b) and in an urban environment (c,d).

The deviation data of the center coordinates of the drone and the landing icon after the landing are derived and sorted into a table, displayed in Table 4. It is evident that the algorithm presented in this paper effectively guides the drone to achieve a precise landing. 

In view of the fact that the coordinate position of the camera of the drone is default and difficult to determine in the simulation environment, the camera coordinate error calibrated by the calibration method may cause a small center deviation. However, the overall navigation algorithm can meet the accuracy requirements of UAV landing. In the future, the UAV camera parameters will be accurately set in the real environment, and then the UAV landing test will be carried out.

## 5. Conclusions

(1) This paper proposes a UAV landing guidance algorithm, Land-YOLO. The algorithm has three improvements on the original model of YOLOv8n: replacing the standard convolution with GhostConv, improving the CSP structure with Pconv, reconstructing the neck network, and introducing the BiFPN module.

(2) In order to evaluate the effectiveness of the improved method, this paper makes a self-made landing-sign data set for the ablation experiment and the comparison experiment. The experimental outcomes indicate that the enhanced algorithm has achieved improvements in both recognition accuracy and reduced computational complexity. Compared with the original YOLO v8 n model, the Land-YOLO algorithm reduces the memory usage and the number of floating-point operations per second by 42.8% and 32.4%, respectively, which is conducive to deployment to the mobile terminal of the drone.

The simulation results show that the Land-YOLO algorithm can successfully guide the UAV to land accurately, which proves that the algorithm has certain reliability and can be used in practical applications.

(3) To evaluate the performance of the algorithm implemented in the UAV terminal, this paper constructs two simulation platforms for UAV flight visualization simulation.

(4) This study mainly verifies the effectiveness of the algorithm through simulation experiments, but in practical applications, actual flight tests are still needed to verify the practicability and reliability of the algorithm. In the future, landing tests of UAV will be carried out in the actual environment, and the algorithm will be further optimized and improved according to the test results.

## Figures and Tables

**Figure 1 sensors-25-00884-f001:**
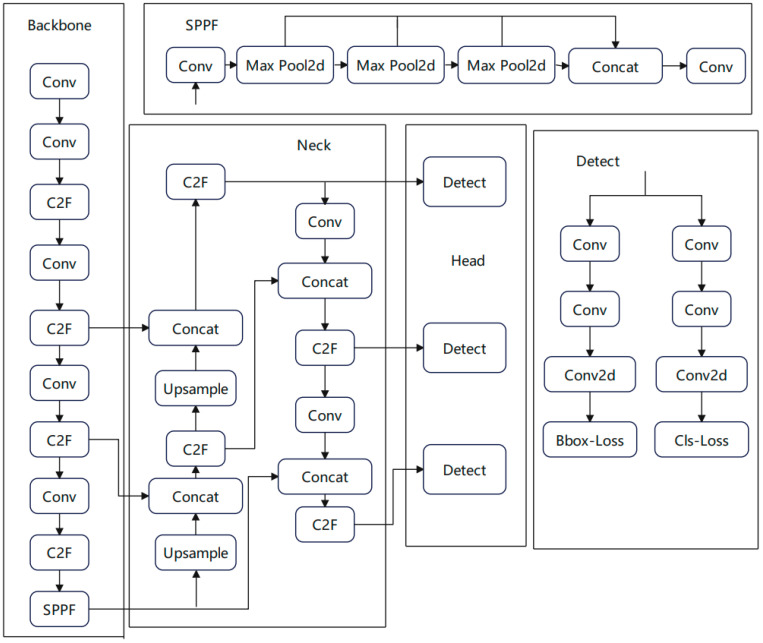
YOLOv8n network structure diagram.

**Figure 2 sensors-25-00884-f002:**
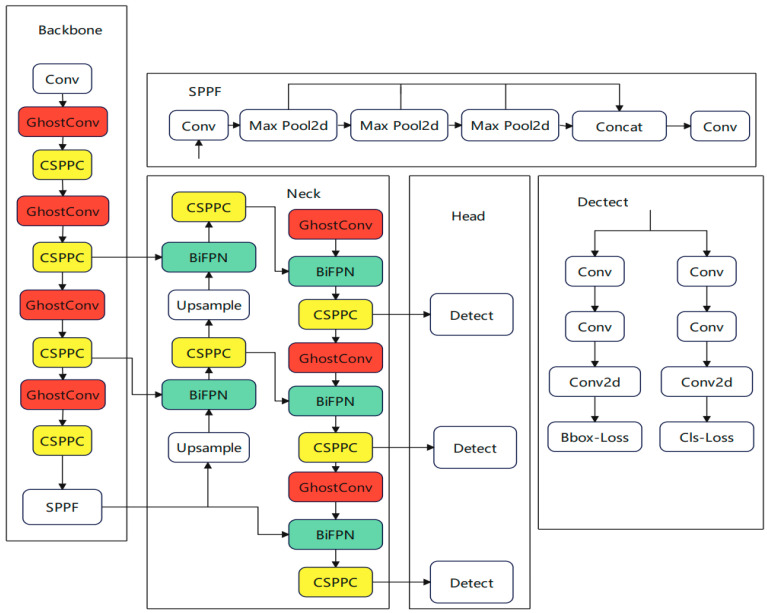
Land-YOLO network structure diagram.

**Figure 3 sensors-25-00884-f003:**
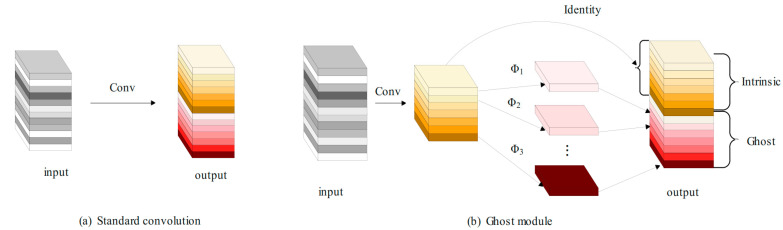
GhostConv structure diagram.

**Figure 4 sensors-25-00884-f004:**
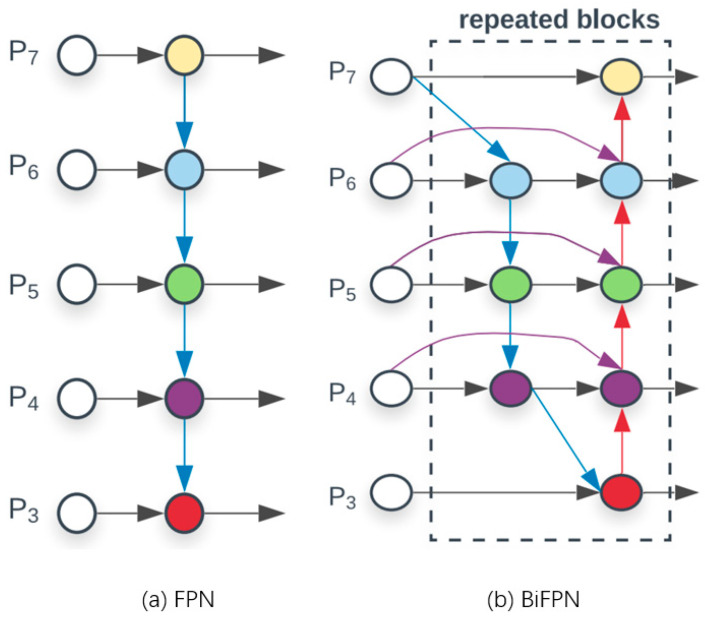
Comparison of FPN and BiFPN structures.

**Figure 5 sensors-25-00884-f005:**
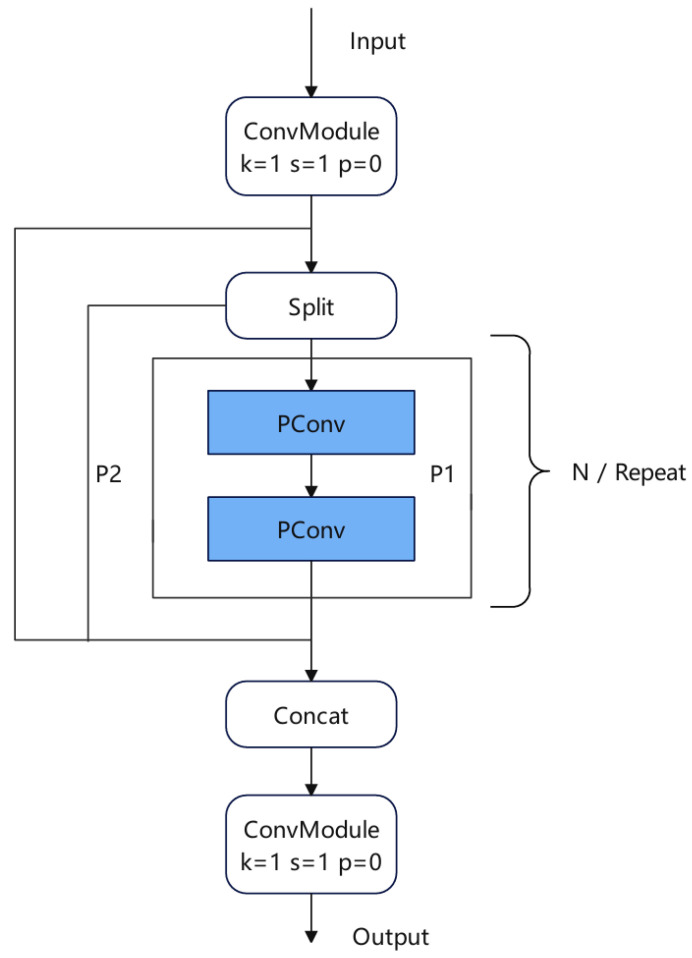
CSPPC structure.

**Figure 6 sensors-25-00884-f006:**
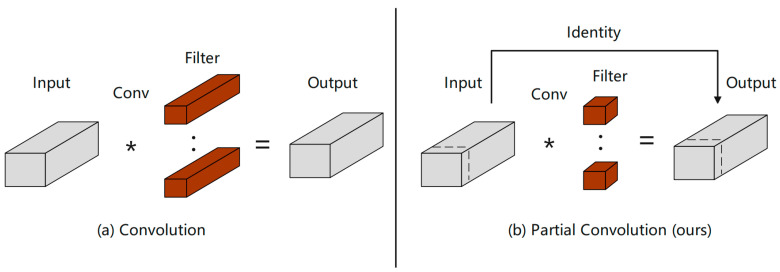
Conventional convolution and partial convolution comparison.

**Figure 7 sensors-25-00884-f007:**
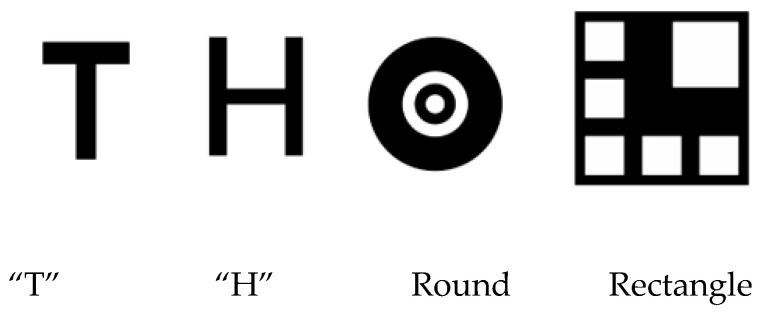
Various landing markings.

**Figure 8 sensors-25-00884-f008:**
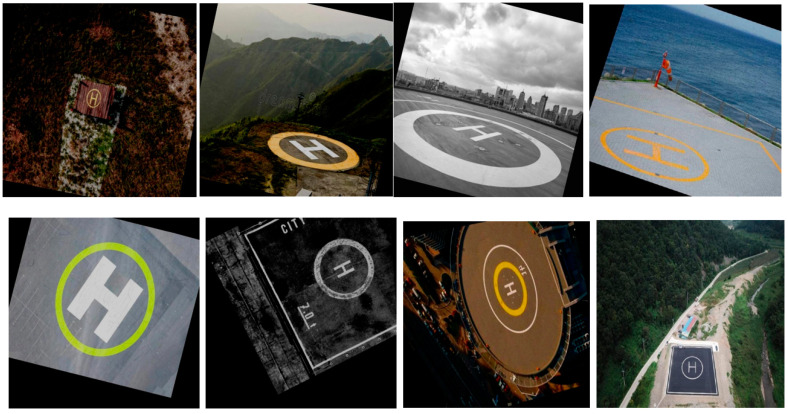
Landing annotation data set.

**Figure 9 sensors-25-00884-f009:**
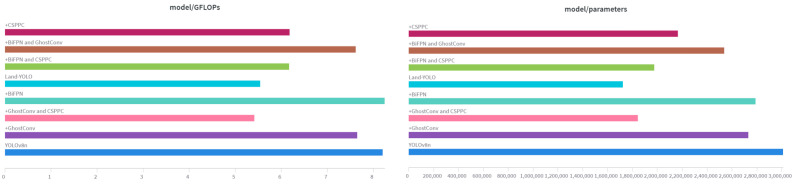
GLOPS and Parameters.

**Figure 10 sensors-25-00884-f010:**
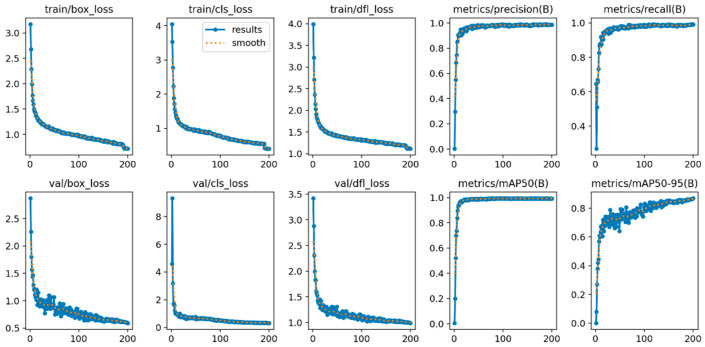
Land-YOLO indicators.

**Figure 11 sensors-25-00884-f011:**
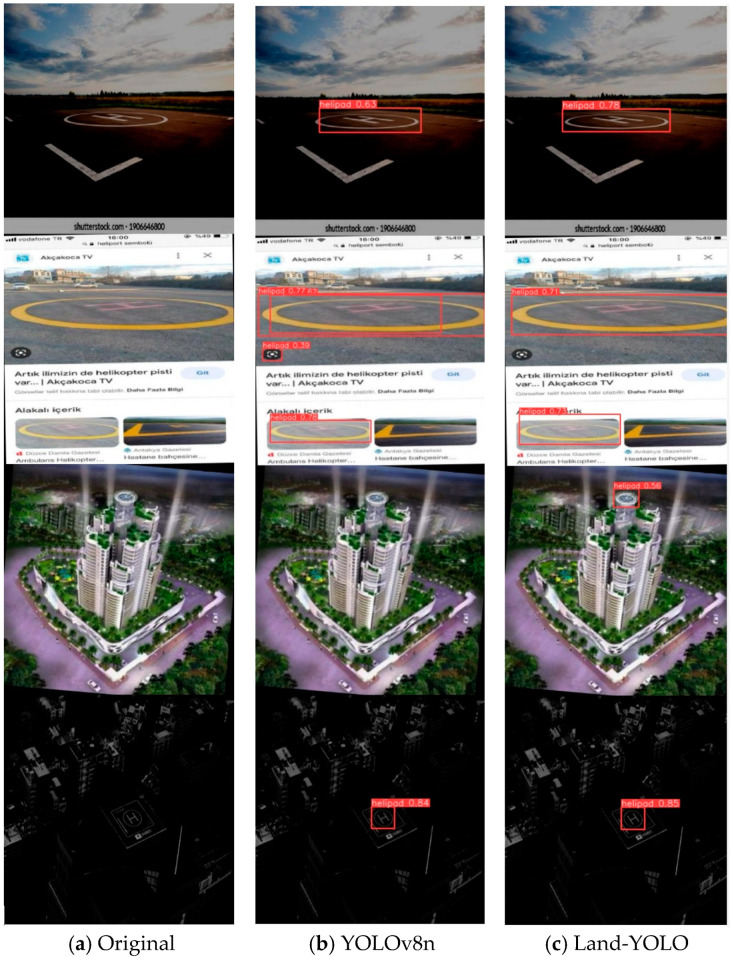
Comparison of detection results before and after improvement.

**Figure 12 sensors-25-00884-f012:**
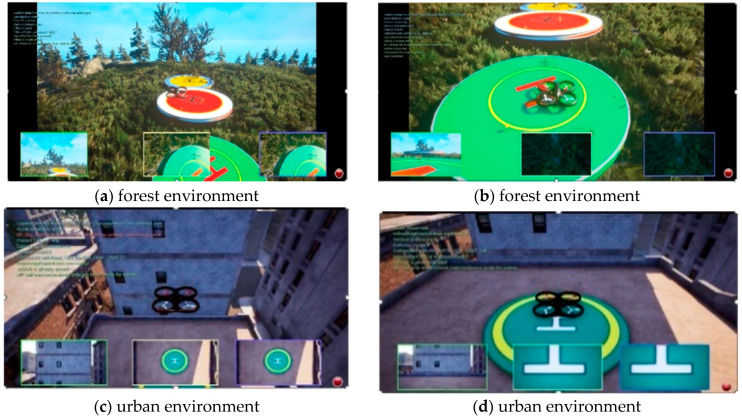
UAV simulation experiment.

**Table 1 sensors-25-00884-t001:** The hyperparameters of the model.

Parameter	Information
imgsz	640 × 640
optimize	SGD
batchsize	16
lr0	0.01
momentu	0.937
epoch	200

**Table 2 sensors-25-00884-t002:** Experimental parameters of training process. Ablation experiment.

Algorithm	P/%	R/%	mAP50/%	GFLOPs	Parmas/M
YOLOv8n	97.87	98.78	98.32	8.194	3.01
+CSPPC	98.17	98.80	98.57	6.177	2.16
+BiFPN	98.94	98.95	99.10	8.24	2.78
+GhostConv	97.98	98.87	98.34	7.647	2.73
+GhostConv and CSPPC	98.83	98.35	98.45	5.41	1.84
+BiFPN and CSPPC	98.27	99.12	98.56	6.166	1.97
+BiFPN and GhostConv	98.86	98.93	99.01	7.614	2.53
Land-YOLO	99.27	99.03	99.23	5.539	1.72

**Table 3 sensors-25-00884-t003:** Experimental results comparison.

Algorithm	P/%	R/%	mAP50/%	GFLOPs	Parmas/M
Faster-RCNN	95.34	95.81	95.45	370.2	137.1
RT-DETR [25]	99.01	98.99	99.02	57.0	19.8
Mask-RCNN [26]	96.12	96.12	96.56	123.0	44.0
YOLOv5n [17]	97.20	98.23	98.21	7.17	2.5
YOLOv8n	97.87	98.78	98.11	8.2	3.0
YOLOv7—tiny [27]	98.87	98.59	98.32	13.2	6.0
YOLOv10n [28]	98.26	98.95	99.12	6.7	2.3
Land-YOLO	99.27	99.03	99.23	5.539	1.72

**Table 4 sensors-25-00884-t004:** The static landing accuracy and mean in the simulation experiment.

Experimental Sequence	*X*-Axis Landing Accuracy (m)	*Y*-Axis Landing Accuracy (m)
1	0.09	0.10
2	0.11	0.10
3	0.09	0.08
Mean value	0.09	0.09

## Data Availability

Data are contained within the article.

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
