# Peer review of "Lightweight UAV Landing Model Based on Visual Positioning"

_sensors, 2025, doi:10.3390/s25030884_

Round 1

Reviewer 1 Report

Comments and Suggestions for Authors

In this paper, a lightweight UAV-guided landing algorithm named Land-YOLO is proposed, which is based on the YOLOv8n model, and improves the accuracy of landing sign recognition through a series of improvement measures, while significantly reducing the memory usage and computational load of the model. The article has some scientific significance, but there are still some problems that need to be improved.

1. The article is not innovative enough, what is the specific innovation point of this article?

2. The logic of the article is confusing, and the grammatical errors are serious, so it is recommended to revise it.

3. The format of the cited references in the article is incorrect, and it is recommended to correct it uniformly.

4. The method proposed in this paper has limited detection ability for small targets, and it may be difficult to detect particularly small targets or long-distance targets even if the BiFPN module is used. It is recommended to increase the layout of subtle features of the target point.

5. It is recommended to extend to dynamic scenes, where the current work is mainly for static landing sites.

Author Response

Thank you for your thoughtful review of this manuscript. We have carefully considered your feedback and have made revisions accordingly. Below, you will find responses to each of your comments.

Comments 1:  The article is not innovative enough, what is the specific innovation point of this article?

Response 1: We acknowledge that the article may not be prominent enough in expressing innovation points. The specific innovation of this paper lies in :

Custom Dataset and Evaluation: In the absence of a public authoritative landing sign dataset, we have created a custom dataset tailored to the needs of our study.

Proposed Land-YOLO Algorithm: We propose the Land-YOLO algorithm, which improves upon the baselineYOLOv8n model by incorporating GhostConv to replace standard convolution, refining the CSP structurewith Pcony, and reconstructing the neck network by introducing BiFPN. These modifications enhance themodel's performance while reducing computational complexity.

UAV Flight Simulation: To further validate the practical application of our algorithm, we have constructed simulation platforms for UAV flight visualization, providing concrete evidence of the algorithm's performance in real-world scenarios.

This paper contributes the above three innovative ideas and research structures, including data set construction, model improvement and platform simulation. Therefore, this paper has certain innovation in the target detection algorithm of UAV autonomous landing.

Comments 2:The logic of the article is confusing, and the grammatical errors are serious, so it is recommended to revise it.

Response 2: Thanks to the review teacher 's correction. We will carefully revise the article to improve its logical structure and grammatical accuracy. At the same time, we will comprehensively check and correct the grammatical errors in the article to improve the overall readability of the article.

Comments 3:  The format of the cited references in the article is incorrect, and it is recommended to correct it uniformly.

Response 3: We have revised the references in the article according to the unified citation format to ensure that it meets the journal 's submission requirements.

Comments 4:  The method proposed in this paper has limited detection ability for small targets, and it may be difficult to detect particularly small targets or long-distance targets even if the BiFPN module is used. It is recommended to increase the layout of subtle features of the target point.

Response 4: We agree with your point of view, but in this study, we focus on the lightweight improvement of the algorithm. The use of BIFPN is to compensate for the accuracy loss and small target detection problems caused by the lightweight process. At the same time, the algorithm only detects a landing sign and does not contain a variety of targets. The integrity of the data set makes the trained model quite effective in the experiment. As shown in Figure 11 of the paper, small targets can still be detected.

Comments 5:  It is recommended to extend to dynamic scenes, where the current work is mainly for static landing sites.

Response 5: Thank you for pointing this out. In the 4.7 chapter of this paper, we deploy the trained model in a virtual drone and test it in a simulation environment, which is a dynamic scenario. In the future, the landing test of UAV will be carried out in the actual environment, and the algorithm will be further optimized and improved according to the test results.

Thank you once again for your valuable feedback on this manuscript. Your comments have been instrumental in improving the quality of the manuscript. We appreciate your time and effort in reviewing our submission.

Reviewer 2 Report

Comments and Suggestions for Authors

This manuscript proposes a Land-YOLO lightweight UAV guided landing algorithm based on the YOLOv8n model to improve landing accuracy and reduce computational consumption. Firstly, ghost conv is used to replace the standard convolution in the Backbone component to reduce the calculation amount. Secondly, Partial Conv (PConv) is introduced to enhance the cross stage partial (CSP) structure in the Neck component to reduce computing and memory access. Finally, a bi-directional feature pyramid network (BiFPN) is employed to enhance the accuracy of the model recognition landing mark. The structured clearly but lacks standardization in expression. Moreover, some meaningful improvements need to be considered.

Main Comments:

1.      This manuscript focuses on visual detection; however, a guided landing algorithm typically involves guidance law and control algorithm. This implies that the precision of UAV landing cannot be solely reflected through detection accuracy.

2.      The algorithm is primarily based on existing techniques, and the innovation is not highlighted enough.

Minor Comments:

1、The introduction section lacks a summary of the contributions of this study.

2、In (1) and (2), the definitions of rs, rc are lacked, and the derivation is not clear. If they represent the calculation amount, it means that ghost conv cannot reduce the amount of calculation required.

3、This paper lacks standardization, for example, a) Abbreviations of technical terms are not fully expanded when they first appear, such as UAV, CSP, and PBVS. b) The concepts of Input, deep network, shallow network are not reflected in Figure 1. The title of Figure 3 does not match the content. The processing P1 and P2 are lack in Figure 5. Figure 10 is not clear enough. c) The physical meaning of the parameters in Table 1 are not explained in the section 4.2.

4、The PartialConv achieves fast and efficient operation by applying filters on only a few input channels. Whether the channel selection is manual or automatic?

The references cited are outdated and lack some cutting-edge researches.

Author Response

Thank you for your thoughtful review of this manuscript. We have carefully considered your feedback and have made revisions accordingly. Below, you will find responses to each of your comments.

Main Comments:

Comments 1: This manuscript focuses on visual detection; however, a guided landing algorithm typically involves guidance law and control algorithm. This implies that the precision of UAV landing cannot be solely reflected through detection accuracy.

Response 1: Thank you for your valuable comment. You are correct in pointing out that a guided landing algorithm not only relies on visual detection but also incorporates guidance law and control algorithm to ensure precise UAV landing. In our study, while the primary focus is on enhancing the visual detection aspect of the landing process, we acknowledge the importance of the guidance and control algorithms. In fact, the ultimate goal of improving detection accuracy is to provide more reliable and accurate input for the subsequent guidance and control systems, thereby enhancing the overall precision of UAV landing. We have conducted simulations in the AirSim environment, where the positioning information and pose estimation during the UAV's landing process are updated based on the improved YOLOv8 visual guidance landing algorithm. These simulations help us indirectly assess the combined performance of detection, guidance, and control, albeit in a simplified manner. However, to fully evaluate the integrated system's performance, a more comprehensive experimental setup involving actual UAV hardware and control algorithms would be necessary, which we plan to pursue in future work.

Comments 2: The algorithm is primarily based on existing techniques, and the innovation is not highlighted enough.

Response 2: We acknowledge that the article may not be prominent enough in expressing innovation points. The specific innovation of this paper lies in :

Custom Dataset and Evaluation:   In the absence of a public authoritative landing sign dataset, we have created a custom dataset tailored to the needs of our study. 

Proposed Land-YOLO Algorithm:  We propose the Land-YOLO algorithm, which improves upon the baselineYOLOv8n model by incorporating GhostConv to replace standard convolution, refining the CSP structurewith Pcony, and reconstructing the neck network by introducing BiFPN. These modifications enhance themodel's performance while reducing computational complexity. 

UAV Flight Simulation: To further validate the practical application of our algorithm, we have constructed simulation platforms for UAV flight visualization, providing concrete evidence of the algorithm's performance in real-world scenarios.

These innovations collectively contribute to the advancement of UAV autonomous landing technology, particularly in improving detection accuracy and reducing computational consumption.

Minor Comments:

Comments 1:  The introduction section lacks a summary of the contributions of this study. 

Response 1:  In order to improve the thesis, we have added a paragraph at the end of the introduction to clearly outline the main contributions of this study.

Comments 2: In (1) and (2), the definitions of rsrc are lacked, and the derivation is not clear. If they represent the calculation amount, it means that ghost conv cannot reduce the amount of calculation required.

Response 2: In order to clarify this point, we will supplement the definitions of rs and rc in the paper and explain the derivation process in detail. Specifically, rs represents the ratio of the amount of calculation through the Ghost module to the amount of calculation through the ordinary convolution, while rc represents the ratio of the number of parameters through the Ghost module to the number of parameters through the ordinary convolution.

Comments 3: This paper lacks standardization, for example, a) Abbreviations of technical terms are not fully expanded when they first appear, such as UAV, CSP, and PBVS. b) The concepts of Input, deep network, shallow network are not reflected in Figure 1. The title of Figure 3 does not match the content. The processing P1 and Pare lack in Figure 5. Figure 10 is not clear enough. c) The physical meaning of the parameters in Table 1 are not explained in the section 4.2.

Response 3:  Thank you very much for your detailed correction. In order to improve the readability and standardization of the paper, abbreviations and image problems have been improved.

Comments 4:  The PartialConv achieves fast and efficient operation by applying filters on only a few input channels. Whether the channel selection is manual or automatic?

Response 4:  Channel selection here is done automatically. The design principle of PartialConv is based on the self-learning ability of the deep learning model. During the training process, the model automatically learns and selects which channels are more important for specific tasks ( such as target detection ), so that only these key channels are filtered.

Some references have been updated.

Reviewer 3 Report

Comments and Suggestions for Authors

1.Recently autonomous landing technology is the cornerstone of the UAV's critical landing phase during flight. The enhancement of the precision of UAV landings is a significant prospective research direction. In the article the authors proposed a Land-YOLO lightweight UAV guided landing algorithm based on the YOLOv8 n model.

2.The Land-YOLO lightweight UAV guided landing algorithm consists of the following main parts:

Part 1. GhostConv replaces standard convolutions in the backbone network, leveraging existing feature maps to create additional "ghost" feature maps via low-cost linear transformations, thereby lightening the network structure.

Part 2. The CSP structure of the neck network is enhanced by incorporating the PartialConv structure.

Part 3.The Bi-directional Feature Pyramid Network ( BiFPN ) module is introduced, and the accuracy and average accuracy of the model recognition landing mark are improved through the bidirectional feature fusion and weighted fusion mechanism.

3.The experimental results are described at a very good level. The results show that the proposed algorithm (method) has sufficiently high effectiveness.

4.In general, the all article parts are prepared at a good levels. The article contains an interesting and useful (for the theoretical viewpoint and from the practical viewpoint) new algorithmic suggestions. The article can be accepted (after minor revisions).

5.Some suggested improvements are as follows:

5.1.To add a generalized scheme a structure of the research (in introduction).

5.2.To add the conclusion section (including a description of the future research directions).

Author Response

Thank you for your thoughtful review of this manuscript. We have carefully considered your feedback and have made revisions accordingly. Below, you will find responses to each of your comments.

Comments 1:  5.1.To add a generalized scheme a structure of the research (in introduction).

Response 1:Thank you for pointing out this, I have added a paragraph in the introduction to show the research structure more intuitively.

Comments 2:  5.2.To add the conclusion section (including a description of the future research directions).

Response 2:Thank you for pointing out this point. I have added a paragraph in the conclusion to show the shortcomings of this scheme and describe the future research direction.

Round 2

Reviewer 1 Report

Comments and Suggestions for Authors

This paper proposes a lightweight unmanned aerial vehicle (UAV) guided landing algorithm named Land-YOLO, which improves the accuracy of UAV landing. The paper has certain scientific significance, but there are still some problems . 1. The detection accuracy of the algorithm proposed in the paper may need to be further improved in scenarios with large elevation changes, long visual distance and complex background, and it is suggested to add more data sets.

2. If there are multiple subimages, you are advised to number them accordingly.

3. The information given in Table 1 lacks corresponding units, which is not conducive to understanding the content of the article

4. As shown in Figure 10, there is only one subgraph with annotations, and whether the annotations of other subgraphs are the same.

Author Response

Thank you for your thoughtful review of this manuscript. We have carefully considered your feedback and have made revisions accordingly. Below, you will find responses to each of your comments.

Comments 1: The detection accuracy of the algorithm proposed in the paper may need to be further improved in scenarios with large elevation changes, long visual distance and complex background, and it is suggested to add more data sets.

Response 1: Thank you for this valuable comment.We agree with your point of view, but in this paper, we focus on the lightweight of the algorithm, and the improvement of the accuracy rate is the secondary research focus. The model calculation amount and model size of the algorithm proposed in this study are reduced by 42.8 % and 32.4 % respectively, and the accuracy rate is also improved to 0.91 % compared with the original algorithm. For its special scene, the detection accuracy of the algorithm may not be as high as that of the ordinary landing scene, but it also has an accuracy rate of about 80 %. At the same time, it can also identify the target that the original algorithm cannot identify ( as shown in Figure 11 ), which is a great improvement in the accuracy of the algorithm. How to further improve the performance of the algorithm in these complex scenarios without increasing the amount of calculation is our subsequent research direction.

Comments 2: If there are multiple subimages, you are advised to number them accordingly.

Response 2: Thank you for this valuable comment.We have numbered all the sub-images in the paper ( Figure 3, Figure 4, Figure 6, Figure 7, Figure 9, Figure 12 ) so that readers can more easily understand and track the image information in the paper.

Comments 3: The information given in Table 1 lacks corresponding units, which is not conducive to understanding the content of the article.

Response 3: Thank you for this valuable comment.All the data in Table 1 usually have no specific physical unit in the deep learning model, but exist as a hyperparameter that controls the training process of the model, usually an integer or decimal.

Comments 4: As shown in Figure 10, there is only one subgraph with annotations, and whether the annotations of other subgraphs are the same.

Response 4: Thank you for this valuable comment. We modified and explained this in detail in Part 4.6 ( page 10, line 318-323 ). In the revised version, all subgraphs are properly annotated and explained to ensure that readers can more fully understand what Figure 10 shows and how they support our findings.